# Artifact as a Node of Heterogeneous Relationships: A Study with Traditional Natural Packaging in Cooking and Food Preparation Practices in Antioquia, Colombia

Carlos Mario Gutiérrez-Aguilar [1],*[iD], Maria Isabel Giraldo Vásquez [2], Juan Pablo Parra Arcila [1], Javier Ernesto Castrillón Forero [1], Mariana Ruiz Restrepo [1] and Alvaro David Monterroza-Rios [1][iD]

1 Departamento de Diseño, Facultad de Artes y Humanidades, Instituto Tecnológico Metropolitano, Calle 73 76A-354, Medellín 050034, Colombia
2 Fundación Universitaria Bellas Artes, Medellín 050034, Colombia
* Correspondence: carlosgutierrez@itm.edu.co

**Abstract:** This article studies natural food packaging as enabling artifacts of the traditional material culture of Antioquia in Colombia. For this purpose, we consider artifacts as objective nodes that combine design and use intentions, functions, materials, histories, artifactual lineages, and cooperative relationships that stabilize ritualized practices of a human group. We take the example of natural packaging as artifacts that enablers and stabilizers of traditional cooking and food preparation practices. Natural packaging materials are here assumed to be leaves having some favorable property to contain a food product. After providing a theoretical reflection, we analyze the data collected from fieldwork we conducted in two towns in Antioquia, Colombia (Santa Fe de Antioquia and Amagá), as well as from an interview with an expert in the field. Finally, we show that it is possible to postulate an analysis under a relational ontological description of a traditional practice with conceptual categories of the philosophy of technology.

**Keywords:** philosophy of technology; technical artifacts; function theory; natural food packages; material culture





## 1. Introduction

Material culture studies [1,2] have become a knowledge problem of particular interest to various disciplines. Since its emergence in the field of archaeology (where it is regarded as a pivotal issue), it has expanded to areas such as anthropology, philosophy, history, language sciences, communication, and industrial design. Material culture as a problem has centered on specific objects of knowledge, including materiality as a feature of culture, musealization, civilizations, and social groups, and the uses of language. One of the main lines of theoretical reflection and research on the matter focuses on the concept of artifact: how to properly define it, what its main characteristics are, what distinguishes it from other forms of human production, and what the intentions of agents producing it are, among other issues.

Within this broad category, there is an interest in investigating the artifacts that mediate foodways both at the community and domestic levels. Although there are interesting publications on the subject (particularly from the perspective of the archaeology of food), very few have addressed the vertical role of natural food packages in the development of some key functions in food consumption. Now, when it comes to natural food packaging using organic materials—a practice in which the organic material is not, or hardly, transformed—this lack of scholarly production is even more pronounced.

Various studies in the field have been conducted for taxonomic purposes. However, although significant, these exercises have overlooked aspects related to material culture. For example, Colombian botanist Santiago Díaz Piedrahita conducted research between

1975 and 1977 entitled *Las hojas de plantas como envolturas de los alimentos* (Leaves as food wrappers) [3], whose reissue was made possible thanks to a project by the Ministry of Culture of Colombia with its *Biblioteca Básica de Cocinas Tradicionales* (Basic Library of Traditional Cuisines). A significant contribution of such study is the classification of natural food packages and wrappers according to their purpose in food preparation and consumption: leaves for wrapping raw food, leaves for cooking, leaves for processed foods, and leaves for food packaging and protection [3].

According to the extensive information provided by Díaz Piedrahita [3], natural food packages, and particularly leaves, have played a crucial role in the foodways of partially industrialized or non-industrialized communities. This paper argues that those natural packages are configured as significant artifacts [4] produced by participatory practices of sense-making between agents [5] given they are nodes that combine design and use intentions, functions, materials, histories, artifactual lineages, and cooperative relationships that stabilize ritualized practices associated with the process of food, and thanks to this condition they are sustained as important elements of the material culture of some societies.

We will first provide an overview of material culture to justify this thesis. Then, we will explore the archaeology of food to identify some conceptual frameworks on food as a vestige of material culture. Finally, we will address the concept of an artifact based on some theories of intentional agency to delimit the objective functions of natural food packages.

We will try to demonstrate that the use of a qualitative case study close to material culture studies could enrich or even resignify the philosophical analyses of technical artifacts that have been developed by the Anglo-Saxon philosophy of technology. In addition, the development of a theoretical framework of networks of practices can give greater theoretical solidity to qualitative approaches to material culture studies.

## 2. Food and Material Culture

According to Gessner, Nandi, and Schwarz-Bierschenk [6], there have been dominant theoretical approaches to material culture, each related to one of three fundamental concepts in the field: objects, things, and materiality.

The object-based approach argues that objects are endowed with a privileged cultural value because they are determined by collective agreements and previously defined purposes [6]. This privileged position implies an ascription of meanings to objects, which would avoid a possible polysemy—understood as the range of changing significations that objects themselves could have.

The thing-based approach, whose most prominent exponent is Arjun Appadurai [7], states that things can change their meanings within contextualized social relations. This approach departs from the object-based perspective in that it does not depend on the unchanging designators of objects and claims that things are susceptible to having unstable meanings depending on their spatiotemporal position.

Lastly, the materiality-based approach abandons both the meaning-centered and relational explanations to favor an interpretive stance. Therefore, its premise is based on the use that is given not only to artifacts but to other systems of signs such as writing [6].

In our concept, we adopt a hybrid position in which the relationship between objects and their meanings is not predetermined or preconceived, but rather emerges in the practices themselves between human agents and stable material referents (artifacts), since they enable the ritualization of actions in which meanings are concretized interactively [8]. In this sense, the meanings do not have an independent existence, but rather emerge in the practices that occur among human individuals among themselves, with and through the artifacts of material culture. Therefore, we understand material culture as an environment of practices that enable and support significant practices that sustain identities (individual and collective). In addition, it is consolidated as a repository of collective memory that is inherited by each new generation [5].

Natural packaging has been consolidated as a family of significant traditional artifacts that, due to their material properties and the eating practices they support, have become the relevant object of study for this research.

In the next section, it will be intuited that, for the present study, the applicable approach is based on practices, since it is suggested that natural packages have qualities highly valued by a group that depends on their material properties. This would imply thinking that the meaning of natural packaging would change if its property of being a food container is eliminated.

Earlier studies into the archaeology of food focused on humans' need to feed to survive. Afterward, the authors concentrated on analyzing different forms of food consumption and their relationship to feasting and celebrations. Recently, attention has been drawn to food preparation and its connection with external variables identified in the socioeconomic sphere [9,10]. This recent research approach makes it possible to investigate the role of food packages (including those made from organic materials that are not transformed via industrial processes). An important thesis emerges from the above: natural packaging is significant when it stabilizes certain eating practices that depend not only on the actions and intentions of the practitioners, but also on their physical properties in the containment, cooking, and disposal of traditional foods. This will be further discussed later in this paper.

Spataro and Villing [11] address a key issue in understanding the role of packaging in food practices. If food is associated with individuals' subsistence, its preparation and consumption are related to their social and cultural dynamics [11]. Consequently, packages are part of organization strategies around foodways, which, in the case of natural food packages, correspond to practices that, for contextual reasons, have not yet been permeated by industrial production.

A difficulty that arises when studying natural food packages as products of material culture from the perspective of the archaeology of food is the impossibility of identifying vestiges that could be used to replicate theoretical models. This implies that studies in the field only provide indirect notions about social practices related to foodways. To counteract this, it is important to introduce a new approach that allows this phenomenon to be analyzed outside of its historical context. In the next section, we will attempt to address this issue.

## 3. Artifacts as Heterogeneous Relationship Nodes

Artifacts have traditionally been studied by the philosophy of technology from different perspectives: (i) some are affirmed in the idea that they are creations of the mind that materialize human intentions; (ii) others see them as functional devices more like organisms than creations of the mind. In addition, (iii) others theoretical perspectives have also been presented to consider them as dual entities between matter and intentions.

For intentionalist approaches (i), an artifact is an object that it is because it has been created to be precisely that object and not another. Some representatives of this approach include Risto Hilpinen [12,13] and Amie Thomasson [14] and according to them, what distinguishes an artifact from a natural object is not the fact that artifacts have functional properties, since that many natural objects such as organs have functional properties. On the contrary, the fact that distinguishes the artifacts is related to the origin of the functional properties that they present. In artifacts, functional properties depend on the mental states of designers, producers, and users [15].

The basic notion of intentional theories is as follows: an entity is included in the category of artifact «x» when it has been produced with the intention that it falls under that category of similar past artifacts, for this reason, many authors call this approach «historical-intentional» [16] (p. 328). Amie Thomasson states that the ontology of an artifact «is constituted by the mental and intentional contents of its makers» [14] (p. 53), for example, a screwdriver is the result of a human intention to produce an object that belongs to the "screwdriver" class.

Diego Lawler [17] has raised some criticism of this approach because intentions are not fully formed until some event, such as a new artificial class, has been fully realized. Artifacts are an important part of our material culture; they populate our ordinary world, the world we interact with. Lawler considers, based on the contributions of Herbert Simon [18], that an artifact must be seen as the interface between the internal structure of an object and its environment. These interfaces are characterized by a set of regularities, which are based on the very nature of the objects. This set of regularities comprises two types: (1) causal patterns derived from the link between the causal dispositions of the internal structure of the object and how it is manipulated; and (2) links established between the object and an environment of intentional agents and actions (dispositions linked to the construction of plans and goal-oriented activities). Consequently, the nature of the object is determined by both kinds of regularities [17] (p. 601).

On the other hand, there is the perspective of understanding artifacts as «functional objects» (ii), because they have been created and selected by a specific cultural group of users and designers to develop certain functions [15] (p. 2). This approach has its origins in the notion of function in biology and its authors seek to extend it to the artificial realm [19]. On the one hand, there are the defenders of a systemic conception, for example, Robert Cummins [20], whose primary notion of function is the causal contribution to the activity of a system. On the other side are the etiological conceptions, developed by Larry Wright [21], according to which the function is chosen not only based on what the element does but also based on its causal history, that is, from how this element has come to be as it is.

It is evident that there are some similarities between biological and artifactual functions, for example, in the ascription of functions, both consider the physical capacities of the object. It is also noted that the function is justified in terms of the causal history of a certain object, which involves a notion of a historical type that comes from the idea of natural selection for organs or deliberative history in the case of artifacts.

However, while both biological and artifactual functions allude to a causal history, they differ profoundly in the type of history that determines the function. Unlike biological functions, artifactual functions directly involve intentional action. For this reason, the function of the artifacts is partially determined using the object and by the practices and intentional contents of the agents involved in the technical actions. In this way, it can be affirmed that the artifactual functions themselves, their use and meaning, are partially determined by the social environment in which the artifact is immersed. In contrast, biological functions respond to the internal logic of natural selection, involved in their phylogenesis, in which their success depends on their relevance to the survival of the organism. Furthermore, in the creation and interpretation of artifacts, an instance or agent is required to carry out a design, while biological function apparently does not require a designer to create an organism or organ with functions. Some authors even consider that function as a fundamental characteristic of an artifact is not enough or simply false [22,23].

Furthermore, a dual approach (iii) holds that technical artifacts are mental and material entities simultaneously. According to its authors, the physical conceptualization can explain how the artifact works in terms of physical processes, but as a mere physical object, it is not a technical artifact because, without its function, the object loses its status as a technical artifact. On the other hand, intentional conceptualization can explain the function of a technical artifact in terms of what it was made for and relate this function to the realization of human ends. There is a world made up of physical objects that interact with each other through causal connections; also, another world with intentional agents that interact through mental elements such as beliefs, intentions, perceptions, or desires [24]. It seems clear that this approach is inspired by two analogies, the first is that artifacts are subjected to processes of variation and selection like organisms in an ecosystem and the second is that they are dual entities analogous to the mind-body problem of the Cartesian tradition.

According to this structure (Figure 1), an artifact has (1) a physical structure and (2) a context of design and use that reflects the intentions of designers and users. The third

branch would be the «function», which, according to them, would be a bridge to unite said material and mental ontologies [25].

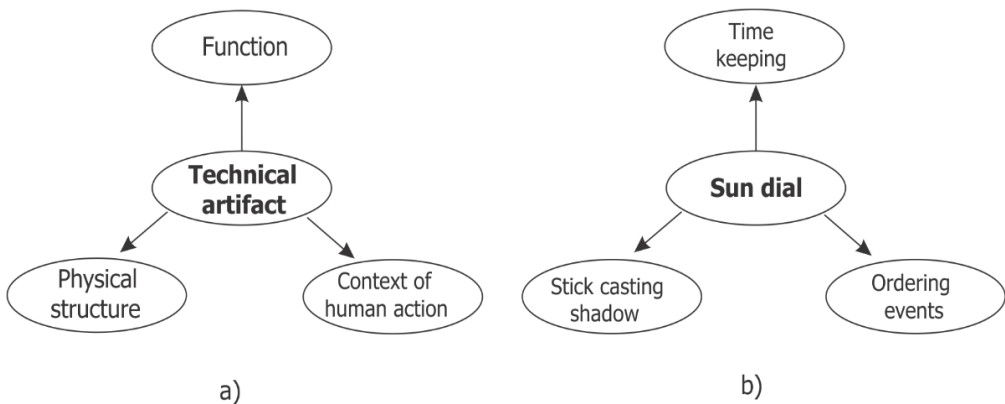

**Figure 1.** (**a**) The dual nature of technical artifacts. (**b**) An example of application. [26] (p. 295). Redrawn by authors.

The dual program provides a persuasive analytical framework for examining issues of artifact design, ontology, and normativity, by their status as hybrid entities in a human context. In addition, the dual approach and its theory of function have a heterogeneous character that links the intentional aspects of the beliefs and goals of the agents with the physical capacity of the artifacts [27] (p. 131). The program has been especially fruitful in bibliographic terms since it has been able to develop, in a few years, a considerable academic production on theories of technical function, the development of concepts around artefactual classes, own functions, and normativity, as well as writing about the effect of social practices on functions. In addition, they have theorized in the field of design engineering and on the study of the moral relevance of artifacts.

However, we consider that it has shortcomings when it comes to explaining the complex dynamics of practices that take place in material culture, especially since artifacts have a wide variety of additional features beyond what the dual program says. In addition to (1) matter, (2) intentions, and (3) technical function, artifacts have other fundamental characteristics that cannot be ignored in an ontological description. For example, (4) the relational character, since all artifacts are necessarily related to human agents and other artifacts, in the networks generated by a material culture; (5) historicity, that is, the fact that an artifact carries a lineage and partially narrates the specific moment and context when it is made and used; (6) the derived complexity, that is, because they form material scaffolding, we delegate part of the social structures to them and establish new horizons of possibilities; or (7) incorporation, since many artifacts become prostheses for individuals who can expand their agent capacities. For this reason, the nature of technical artifacts is neither dual nor triple nor quadruple, their nature is simply heterogeneous [27] (p. 213).

In this sense, we defend a vision of artifact that is complementary to the intentionalist, functionalist, or dual approaches that collect the explanatory advantages of these approaches but that complements the dimensions that are not mentioned in the explanation of practices in material culture. We consider artifacts as objective nodes that combine design and use intentions, functions, materials, histories, artifactual lineages, and cooperative relationships that stabilize ritualized practices of a human group. They are a node of relationships in the complex practices of material culture (Figure 2).

In summary, artifacts are nodes of heterogeneous relationships because they are material elements whose forms are molded through technical agents' designs and action plans (explicit or not). These material pieces fulfill functions (practical or symbolic) relevant to certain human groups which are recognized by their use. In this way, artifacts can only be built within and with a network of previous artifacts (tools, materials, machinery, etc.), knowledge (operational, representational), symbols (forms, tastes, meanings), and

institutions (norms, laws, techniques). Artifacts are «historically» reproduced to support established practices or to address new possibilities. Therefore, any artifact is always in a network with humans and other artifacts, forming a niche of practices that we call «material culture». Every human agent interacts with the environment and with others through the mediation of artifacts (incorporated or not) [27].

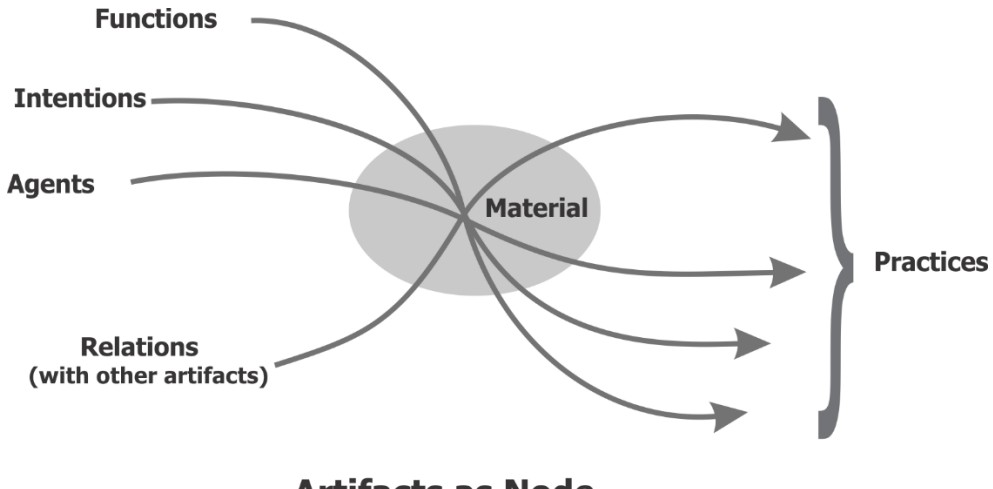

**Figure 2.** Artifact as a node of relationships between intentions, functions, agents, and relationships with other artifacts that enable and ritualize practices in a niche of material culture.

Natural packaging is a type of artifact that enables traditional feeding practices due to its material, and formal and biochemical properties, as well as its availability in characteristic environments, its low cost, and its ability to maintain and ritualize actions. They are, thus, central nodes that support diverse feeding practices and favor participatory sense making encounters [28].

### 4. Fieldwork in Municipalities of Antioquia-Colombia

To verify whether the theoretical foundations outlined in the previous section could be translated into specific practices, we conducted: (1) on-site interviews, combined with direct observation in Santa Fe de Antioquia and Amagá (towns in the department of Antioquia, Colombia), and (2) an interview with an expert in the traditional cuisines of some of the most representative regions in Colombia. In both cases, thematic guides were used to allow participants to provide in-depth reflections on the subject and share their personal experiences and opinions on the use of natural materials for packaging food products.

We selected the El Espinal sector in Santa Fe de Antioquia as the study area because it is one of the places where peasant markets[1] are held. In this sector, we were able to approach some individuals who produce and use natural food packages both commercially and domestically. Although tamale[2] is the most popular food that is wrapped in leaves, other products in this town such as cheese and *fiambre*[3] are also wrapped in these organic materials. A very broad overview of how goods are produced allows us to identify two types of actors in the production chain: (a) those who have control over the entire process (from the harvest of the leaves used for packaging to the commercialization of the finished products) and (b) those who play a specific role within the context of work specialization.

Rubén Darío, a farmer dedicated to growing various types of plants and selling their leaves in Santa Fe de Antioquia, acknowledges that the use of these organic materials to wrap food products has declined significantly. This suggests that technical systems have managed to gain a significant foothold even in areas with a strong rural influence. Rubén Darío explained to us: People used to wrap meat in bijao[4] leaves, but they no longer do that. Leaves were replaced with plastic bags. Meat spoils in plastic bags because of the heat, whereas it retains its freshness in bijao leaves.

The use of materials other than leaves is driven by public policy decisions. According to Fabiola, a small-scale food producer and seller from Amagá, traditional food practices might be lost due to the strict supervision of the local health secretariats and the Instituto Nacional de Vigilancia de Medicamentos y Alimentos (National Institute for Food and Drug Surveillance) in Colombia. This argument is supported by Jainer Grisales, a professor and expert in traditional cuisines at the Colegio Mayor de Antioquia, who argues that these regulatory bodies go against the traditions behind the development of communities and territories—traditions that have become products of material culture. However, to avoid falling into the naturalistic fallacy of granting an attribute of intrinsic goodness to all that is natural, this discussion must be addressed from a different angle. Interestingly, this study is the emergence of an idea peripheral to the central premise: The dynamic between producers who use natural food packages and control bodies suggests a clash of intentions between agents with different purposes.

Liria, a plant grower and tamale maker from Santa Fe de Antioquia provides her insights into this discussion: Using plastics to pack food products implies higher costs. When food is wrapped in plastic, it spoils faster than when wrapped in leaves. In addition, the leaves add a very good flavor to them. Beyond analyzing the direct correspondence of these statements with reality, we identify a belief loaded with intentionality in Liria's discourse. Since every belief is ascribed a propositional attitude, we can assess whether it is true or false. Although we will not delve into this issue, it is worth noting that, in addition to a clash of intentions, there is also a clash of beliefs between producers who use natural food packages and regulatory bodies.

Liria's statement about the flavor that leaves add to the food they contain is consistent with what researchers in the field of traditional cuisine have reported. For instance, in a news story by Caracol Radio, Basilia Murillo, a traditional cook from Cali, stated: *Leaves add a unique flavor to foods, help us rescue our traditions, and are not a source of pollution* [29]. The use of leaves for food wrapping, besides depending on their availability in the environment, incorporates beliefs rooted in consumption habits. Taste and preservation, thus, become key factors to justify the use of these organic materials for food presentation.

In this same vein, Fabiola, a peasant from Santa Fe de Antioquia, explains: ( . . . ) the leaf not only protects the tamale so that it can be cooked and its contents can remain inside but also adds flavor to it. This acknowledges both the physical functions of the artifact (which describe its purpose) and the perceptions about it. Leaves are presumed to improve the taste of food. This assumption, however, is founded on cognitive, and, therefore, strictly subjective, constructs. Then, how can the discourse on flavor be objectified? Why is it often said that leaves improve this attribute? This objectivity is constructed in an intersubjective fashion. David Linden, a neuroscientist at Johns Hopkins University explains this in an interview for BBC: There are other things that we learn to enjoy. For example, although we are programmed to like sweets, personal preferences are influenced more than anything by individual experience, learning, family, and culture—all the things that make us individuals [30]. From this perspective, intentionality ascribed to flavor is determined by the proximity relationships between agents. The intention to prepare, for example, a tamale wrapped in banana leaves is the result of accepting a collective belief that this is the best way.

In addition to taste, raw material availability is also a key element of analysis. In this regard, Liria explains: These tamales are wrapped in banana leaves because we had no bijao plants. ( . . . ) Bijao leaves are perhaps more popular because they do not require any special preparation other than cleaning them. ( . . . ) Here, we use bijao, banana, plantain, or popocho[5] leaves depending on their availability. She states that using one type of leaf instead of the others is not a big problem. Thanks to the large variety of leaves in the area, they can be employed interchangeably as long as they serve the essential purpose: to produce an artifact to contain food while transferring some of their properties. As reported by Díaz (2012), there are 132 plant species in Colombia whose leaves are used as wrappers.

This wide range of alternatives allows for the continuous development of low-cost and long-lasting processes that are inserted into the material culture of these territories.

The availability of leaves, however, does not ensure a global-scale production. Nor can it be said that this is the ultimate goal of producers. Fabiola, for instance, emphasizes the domestic and artisanal nature of these products (e.g., cheese, butter, tamales, and *fiambres*), which makes them a short shelf life. They must be consumed immediately, as opposed to industrial products, which last longer thanks to the use of preservatives. In addition, the technical means available lead to lower productivity, thus resulting in few commercial exchanges other than local sales.

Regardless of the area within which these products circulate, the function related to transport is also important. Professor Jainer Grisales explains it using the transport of eggs in Cauca as an example. In this region, some materials that undergo moderately complex production processes are employed to protect eggs from cracking. Even in products where packaging materials are also used as wrappers, the intention is to solve a logistical problem. This allows the product to, on the one hand, retain its shape and, on the other hand, be stacked both during transport and when exhibited at the point of sale.

Based on our fieldwork, we have discussed some elements that are considered functions resulting from the intentional attitudes of producers using natural materials for food packaging. The results of this study will allow us to draw some conclusions to support new research processes that further contribute to the body of knowledge on this phenomenon.

## 5. Discussion

As we pointed out above, people (agents) who have been traditionally engaged in food preparation were selected for the interviews and, in this sense, their actions are guided by their knowledge and skills that they have learned through their actions in the same traditional cooking and food preparation practices.

The interviewees point out the advantages of traditional natural packaging over other industrial materials. Thus, the physical, chemical, and organoleptic properties of these packages support specific functions that give rise to culinary practices that have been consolidated for generations. This sets them apart from other types of packages, which do not completely lose their properties when the food is consumed. These latter preserve some characteristics that, in some cases, allow them to be reused and continue to serve their communicative purposes, in addition to the museum value they acquire for collectors.

Cooking and food preparation practices ascribed to natural food packaging can be understood as the volitional act of developing a set of functions, which include:

- Protection during cooking: the intention to ensure that the product acquires and retains the characteristics desired by the agent producing it. Note that, in the previous preparation phases, there are also intentional acts that anticipate the moment the food interacts with the leaf;
- Integration with the product: the intention (at least in some food preparations) to add flavor to the product. As reported in the interviews and other sources of information, this characteristic has a significant value in preserving the tradition of preparing food in this way;
- Product container: the intention to prevent the product from losing its consistency due to the physical forces acting on it;
- Transport solution: the intention to solve the logistical problem of transporting the product in the event it could not be contained in the natural package;
- Stacking: the intention to solve the logistical problem of having an optimal space to stack the products without compromising their shape.

Furthermore, we identified another function that is not associated with the products themselves but with a different practices of agents: the sale of leaves as raw material for third-party manufacturing processes. The agents' purpose here is to maximize profit. Therefore, given their limited production capacity, farmers try to take advantage of the availability of these materials to sell them to third parties.

If we compared this information to that of industrial packaging, there would be at least two more functions that are to be reviewed and would cover the scope of other studies. First, it would be important to investigate whether natural packages are intended to preserve food. It is well known that, in the case of industrial packaging, there is a regulated intention to prevent food products from rapidly spoiling. In this study, however, this intention cannot be confirmed because no supporting evidence was collected. Second, we stress the difficulty in determining whether natural food packages serve a communicative function (common to industrial packages). If they comprised communicative elements, these would not be very evident. To solve this difficulty, this issue should be further studied.

Applying the explanatory framework developed in Figure 2, we can see the map of heterogeneous relationships present in the practice under study and how natural packaging occupies a central place in this network of relationships (Figure 3).

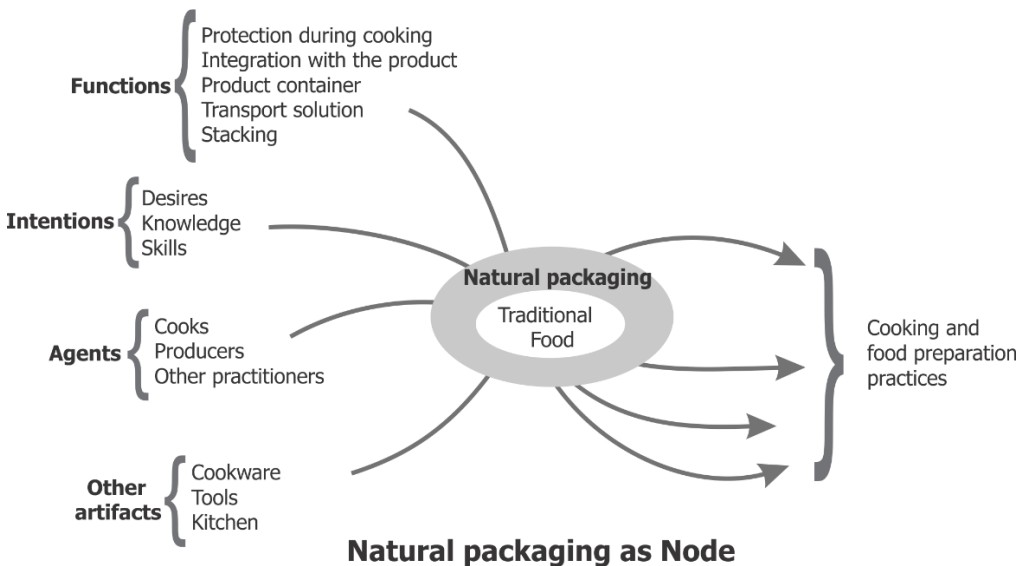

**Figure 3.** Natural packaging that makes traditional cooking and food preparation practices possible.

The intentions mentioned in Figure 3 are linked to the traditional knowledge and skills that the relevant agents (cooks and traditional food producers) learned in the niches of the same practices. The intentions indicated in the analysis are not independent of the practices or the framework that sustains them, such as: natural packaging, traditional ingredients, cooking utensils and tools, and, of course, of the interaction and permanent cooperation with other agents knowledgeable about traditional cooking practices.

In this way, intentions, histories and traditions, and relationships with other human agents, and relationships with a set of artifacts (utensils, tools) and cooking spaces consolidate and reproduce traditional food preparation practices. Within these relationships, as stable and material nodes, are natural packaging.

## 6. Conclusions

This heterogeneous and relational ontology of the artifacts that we propose tries to account for the networks of relationships that an artificial object establishes with the design, use, contextual conditions, and other artifacts. Considering its physical and material structure that redirects causal flows of the physical world.

Natural packaging is a kind of artifact that enables and stabilizes traditional cooking and food preparation practices. The uses and functions also reveal a condition that is unique to natural food packages: these containers are rightly regarded as an integral part of a given product. This implies considering this latter in its entirety; otherwise, we would no longer be speaking of identical expressions. Thus, if tamales were marketed in containers other than natural wrappers, they would no longer be considered such because their meaning

would be lost. In other words, the set of properties that characterize a tamale, including the functions ascribed to its natural package, is what makes it a tamale. This raises a new hypothesis that should be thoroughly discussed in future research.

According to the results of the interviews, natural packaging has physical, biochemical, and organoleptic properties that enable fundamental functions in cooking and food preparation practices in the municipalities of Antioquia (Colombia), especially for cooking protection, integration of food preparation, and product containment. They also facilitate transport and storage for marketing and use in traditional food. In addition, it has the quality of being sustainable because they are renewable in its production and biodegradable if its waste is properly disposed of. All these qualities explain why they have traditionally been constituted as material nodes that support and give meaning to various participatory practices that favor collective sense-making encounters.

We believe that this hybrid work between philosophy of technology and «material culture studies» can be enriching for both disciplinary traditions. On the one hand, understanding practices in a relational way in which the unit of study are the nodes between heterogeneous entities (agents, artifacts, knowledge, lineages, histories, etc.) can contribute to overcome the essentialisms and dualisms that have dominated the analyses in the philosophy of technique by including the interactive, historical, and relational elements that are common in material culture studies. Moreover, an analysis such as the one proposed in this paper can give conceptual precision and a more solid theoretical framework to qualitative descriptions of «material culture studies».

**Author Contributions:** Conceptualization, A.D.M.-R. and M.I.G.V.; methodology, M.I.G.V. and M.R.R.; validation, C.M.G.-A., A.D.M.-R. and J.E.C.F.; formal analysis, C.M.G.-A. and A.D.M.-R.; investigation, M.I.G.V., C.M.G.-A., J.E.C.F., J.P.P.A. and M.R.R.; resources, M.I.G.V. and C.M.G.-A.; writing—original draft preparation, M.I.G.V., J.P.P.A. and A.D.M.-R.; writing—review and editing, C.M.G.-A., A.D.M.-R. and J.E.C.F.; project administration, C.M.G.-A. and M.I.G.V. All authors have read and agreed to the published version of the manuscript.

**Funding:** This research was funded by Instituto Tecnológico Metropolitano, grant number 20207 and the APC was funded by authors.

**Institutional Review Board Statement:** Not applicable.

**Informed Consent Statement:** Not applicable.

**Data Availability Statement:** Not applicable.

**Acknowledgments:** *Oficina de Traducción* ITM provided constructive feedback on a draft in English of this manuscript.

**Conflicts of Interest:** The authors declare no conflict of interest.

## Notes

[1]     Spaces for small and medium-scale producers from rural areas to sell their goods directly to consumers.
[2]     A traditional Mesoamerican dish made of rice- or corn-based dough and usually filled with meat, vegetables, or fruits. Its preparation requires wrapping it in leaves from local plants (e.g., corn husks and banana leaves).
[3]     A dish popular in Antioquia (Colombia) wrapped in banana leaves and generally consisting of rice, chorizo, fried pork rinds, ground meat, hard-boiled or fried egg, a slice of ripe banana, and arepa.
[4]     A plant species (Calathea lutea) that grows in the American tropics. Its leaves are used in some countries to wrap tamales and other food.
[5]     According to the dictionary of the Real Academia de la Lengua Española, it is a low-quality variety of bananas.

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
