# Peer review of "Artifact as a Node of Heterogeneous Relationships: A Study with Traditional Natural Packaging in Cooking and Food Preparation Practices in Antioquia, Colombia"

_philosophies, doi:10.3390/philosophies7050119_

Round 1

Reviewer 1 Report

This article is devoted to the study of the presence of artifacts in people's lives. The article is interesting in the aspect of applying philosophical ideas to the understanding of life practices. However, the article should be strengthened by considering the topic of food as an element of ethnic and cultural identity and intercultural interaction. In this case, this topic acquires a bright philosophical coloring. Attracting sources dedicated to material culture is not enough. The Discussion section needs to be finalized. The authors need to show how their results make it possible to reinterpret existing theories. The conclusion needs to be finalized. Conclusions should be drawn taking into account the philosophical concepts raised by the authors. In its present form, the Conclusion does not correspond to the article on philosophy.

Author Response

Good morning

We have carefully read the suggestions and observations of the reviewers of the article. For this reason, we have made the following changes in specific aspects that are highlighted in the main document. Specifically, these are the changes:

  1. The title and abstract of the article were modified to give greater precision to the objective of the article and its main result.
  2. The mixed analysis that we make between the philosophy of technology and the studies of material culture coming from anthropology was clarified in the introduction (1).
  3. Figures 1 and 2 were redrawn to be clearer in the aspects described and the homogeneity in style and typography. In figure 2, some elements were modified to give more clarity to the main idea being defended.
  4. In the discussion section (5), emphasis was placed on the information provided by the interviewees to adjust the characteristic functions of the cooking and food preparation practices under study, as well as the advantages of traditional natural packaging.
  5. In the discussion section (5), Figure 3 was added, in which the general theoretical framework of Figure 2 is applied to this case. In this way, the relationship between the theoretical part coming from the philosophy of technique with the field work and the qualitative methodologies of material culture studies is clearer. The conceptual categories used were extended to this specific case and how they relate to each other.
  6. Adjustments were made to the conclusions (6) according to the suggestions of the evaluators and the discussion analysis was expanded within the framework of the philosophy of the technique. In other words, it was clarified that the most important results of the article are within the scope of the philosophy. In the same conclusions (6), the hybrid character of the analysis was noted, since it can be understood as a work of philosophy of technique and artifacts, enriched with elements of material culture studies.

Cordially

The authors

Reviewer 2 Report

The article contains a strong thesis on philosophy of technology and material culture, namely the idea that artifacts are constituted in relations of various ontological levels and change over time. To this end, it develops a very interesting analysis of the transformations in traditional food packaging in Colombia. The article carefully develops these two parts, but I suggest to make the intention clearer in the introduction. Perhaps anticipating what appears in the conclusions, it is not clear the link between the two parts: the controversies about the notion of artifact and the controversies about the appropriateness of changing the packaging system. 

Author Response

(The authors gave the same response as above.)

Round 2

Reviewer 1 Report

After finalizing the text, the article looks conceptually designed. The article is interdisciplinary, which from the point of view of philosophy is a successful solution, as it reinforces the theme of the philosophical foundations of everyday life.